# The intergenerational transmission of educational attainment: A closer look at the (interrelated) roles of paternal involvement and genetic inheritance

**Renske Marianne Verweij** *, **Renske Keizer**

Department of Public Administration and Sociology, Erasmus University of Rotterdam, Rotterdam, The Netherlands

* rm.verweij@cbs.nl

**Data Availability Statement:** Data cannot be shared publicly because of the extensive restricted-use data. Information on how to obtain the Add Health data files is available on the Add Health

## Abstract

Numerous studies have documented a strong intergenerational transmission of educational attainment. In explaining this transmission, separate fields of research have studied separate mechanisms. To obtain a more complete understanding, the current study integrates insights from the fields of behavioural sciences and genetics and examines the extent to which paternal involvement and children's polygenic score (PGS) are unique underlying mechanisms, correlate with each other, and/or act as important confounders in the intergenerational transmission of fathers' educational attainment. To answer our research questions, we use rich data from The National Longitudinal Study of Adolescent to Adult Health (n = 4,579). Firstly, results from our mediation analyses showed a significant association between fathers' educational attainment and children's educational attainment (0.303). This association is for about 4 per cent accounted for by paternal involvement, whereas a much larger share, 21 per cent, is accounted for by children's education PGS. Secondly, our results showed that these genetic and behavioural factors are significantly correlated with each other (correlations between 0.06 and 0.09). Thirdly, we found support for genetic confounding, as adding children's education PGS to the model reduced the association between paternal involvement and children's educational attainment by 11 per cent. Fourthly, evidence for social confounding was almost negligible (the association between child's education PGS and educational attainment was only reduced by half of a per cent). Our findings highlight the importance of integrating insights and data from multiple disciplines in understanding the mechanisms underlying the intergenerational transmission of inequality, as our study reveals that behavioural and genetic influences overlap, correlate, and confound each other as mechanisms underlying this transmission.

## Introduction

Individuals with highly educated parents generally achieve better outcomes in school than individuals with lower educated parents [1]. For example, during primary and secondary

website (http://www.cpc.unc.edu/addhealth). The data underlying the results presented in the study are available from CPC Data Portal (https://data. cpc.unc.edu/projects/2/view). Restricted-use data will be distributed only to certified researchers who commit themselves to maintaining limited access. The authors had no special access privileges, and other researchers will be able to access the data in the same manner.

**Funding:** The present study was supported by a grant from the Netherlands Organization for Scientific Research to RK (NWO MaGW VIDI; grant no. 452-17-005) (https://www.nwo.nl/en) and by a grant from the European Research Council to RK (ERC StG; grant no. 757210) (https://erc.europa. eu/). Add Health is directed by Robert A. Hummer and funded by the National Institute on Aging cooperative agreements U01 AG071448 (Hummer) and U01AG071450 (Aiello and Hummer) at the University of North Carolina at Chapel Hill. Waves I-V data are from the Add Health Program Project, grant P01 HD319121 (Harris) from Eunice Kennedy Shriver National Institute of Child Health and Human Development (NICHD), with cooperative funding from 23 other federal agencies and foundations The funders had no role in study design, data collection and analysis, decision to publish, or preparation of the manuscript.

**Competing interests:** The authors have declared that no competing interests exist.

school years, children with highly educated parents obtain higher school grades [2], and eventually, children with at least one highly educated parent are more than twice as likely to obtain tertiary education themselves compared to children without highly educated parents [3]. Different research fields focus on different factors in explaining this intergenerational transmission of educational attainment. Research from the field of behavioural sciences aims at explaining this intergenerational transmission by considering, amongst others, characteristics of the family environment, such as parenting practices and the level of social, cultural, and economic capital within families [4, 5]. In contrast, research from the field of genetics focuses on the role of genetic transmissions [6].

With a few exceptions [7–10], studies that have looked at the *interrelated* linkages between behavioural and genetic factors are rare. That such research is rare is very unfortunate; investigating behavioural and genetic factors in isolation most likely overestimates the unique contribution each factor makes. In line with this idea, previous studies have revealed that controlling for genetics reduced the impact of parenting quality [8] and family environment [10] on children's educational achievement, which indicates that part of the assumed social effect might be genetic, so-called 'genetic confounding'. Genetic confounding arises amongst others because the shared genetic influences that parents pass on to their offspring collectively contribute to both increased genetic predisposition for variation in educational attainment as well as the environments that parents design for their children. This might be because traits that are assumed to be environmental, such as family SES, are partly accounted for by genetic factors [7]. Research has also shown the reverse; part of the genetic effect on education might be socially confounded; empirical studies have revealed that the association between parents' and children's education-related genes and children's educational attainment is reduced when parenting practices and family SES are taken into account [9–11].

These findings underscore the importance of taking both behavioural and genetic factors into account to obtain a clear understanding of the mechanisms underlying the intergenerational transmission of educational attainment, which is the aim of the current paper. Our study builds forward on work by Wertz et al (2020). Although these authors did not investigate the intergenerational transmission of educational attainment, they did scrutinize the role of both maternal behaviour and genes in explaining children's educational attainment. Their study revealed that mothers' cognitively stimulating parenting accounted for the effect of mothers' education polygenic score (PGS) on children's educational attainment and that the inclusion of children's education PGS slightly reduced the association between mothers' parenting (cognitive stimulation, warmth, and sensitivity) and children's education attainment [8]. This, amongst others, shows that part of the association between parental genes and children's education is accounted for by the parenting mothers provide to their children. In the current paper, we turn our attention to the role of fathers (whilst controlling for the role mothers play).

Traditionally, studies on the intergenerational transmission of education/socioeconomic status (SES) focused on obtaining information on fathers' educational attainment or profession. As women, particularly in previous decades, often retreated from the labour market after marriage or childbirth, information on mother's educational attainment or profession was not always available for each child. In contrast, information on the role fathers played in parenting was often neglected or overlooked, as mothers were the primary source of information to report on child development. During the last 50 years, however, fathers have become more and more involved in parenting [12, 13]. Although some scholars argue and show that certain roles might still be most prominent amongst solely mothers or solely fathers, the roles of fathers (and mothers alike) are increasingly being expanded [14], which have made fathers and mothers more similar in their roles as caregivers [15]. Even though some scholars construct a

gender-differentiated vision of *how* mothers and fathers parent, there is increasing consensus among scholars that there are little to no differences in how *well* mothers and fathers parent [e.g. 15]. In line with this, and pertaining to educational attainment, studies show that the relationship between parental involvement and academic achievement is similar for mothers and fathers [16].

That said, we do expect to see greater variation in paternal than in maternal involvement. Over the years, fathers in two-parent families have spent more time with their children [17], a pattern especially common among higher educated fathers [18]. After divorce, which is more common amongst lower-educated families [19], fathers often are, or become, less involved in their children's lives, which is especially the case among lower-educated families [20]. This greater variation in father involvement across social strata suggests that father involvement could be an important underlying mechanism of the intergenerational transmission of educational attainment, and thus possible leverage to reduce inequality.

Several studies have investigated the role that paternal involvement plays in children's educational attainment and in the intergenerational transmission of educational attainment [21, 22] Unfortunately, however, these studies did not control for genetic effects. Even though twin studies show that both genetic aspects, as well as the shared environment, are important in accounting for children's educational attainment, the shared environment in these studies remains unmeasured [6, 23–25].

In sum, the current study aims to provide a clearer understanding of the (interrelated) roles that paternal involvement and genes play in the intergenerational transmission of educational attainment. We use children's PGS for educational attainment as our genetic indicator. This PGS is based on a genome-wide association study (GWAS) conducted among 1.1 million individuals [26]. In this GWAS, the association between hundreds of thousands of genetic variants and educational attainments is assessed. These GWAS summary statistics are used to calculate the sum of all risk alleles, weighted by their reported effect sizes. A PGS can therefore be seen as the summary measure of the genetic propensity for a trait based on a large number of genetic variants [27]. The Education PGS has been found to account for about 11–13% of the variation in educational attainment [26]. We will examine whether and to what extent genes as well as father involvement mediate the relationship between father's and child's educational attainment, whether and to what extent these mediators are correlated, and/or act as important confounders in the intergenerational transmission of fathers' educational attainment.

## Theory and hypotheses

### Paternal involvement as mechanism underlying the intergenerational transmission of educational attainment

To build the argument that fathers' involvement in their children's lives is an underlying mechanism for the intergenerational transmission of educational attainment, we would first have to argue and show that (1) fathers' educational attainment is significantly associated with children's educational attainment, that (2) fathers' involvement is significantly associated with children's educational attainment and (3) fathers' educational attainment is significantly associated with fathers' involvement in their children's lives (see Fig 1 top part).

Firstly, numerous studies have revealed that fathers' educational attainment is associated with children's cognitive functioning [28]. Children of highly educated fathers not only show better cognitive functioning but also obtain better school grades throughout their educational career [2] and higher educational levels [3]. For example, in the US less than 20% of the children without tertiary educated parents obtain tertiary education themselves, while around 55% of those with at least one tertiary educated parent do so [3]. Secondly, fathers' involvement in

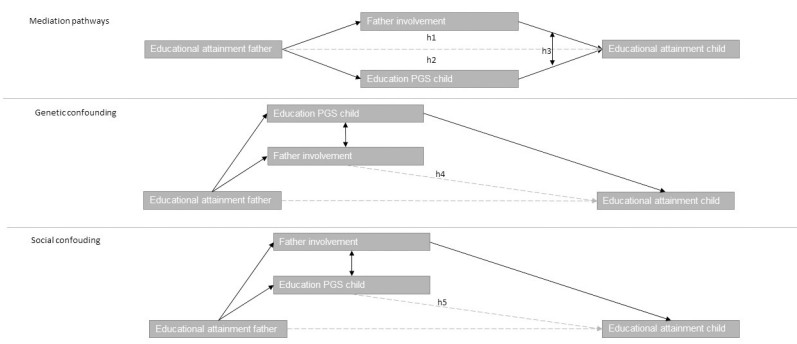

**Fig 1. Graphical presentation of hypotheses 1, 2, and 3: Mediation pathways and correlated effects, hypothesis 4: Genetic confounding and hypothesis 5: Social confounding.**

their children's lives has been argued to positively affect their children's educational attainment. Different dimensions of fathers' involvement might uniquely influence children's educational performances. First of all, and most directly, fathers may influence their children's educational outcomes through their involvement in teaching-related activities such as coaching, helping with homework, communication with school personnel, and active participation in classroom or school activities [21, 29, 30]. The involvement of fathers may benefit children's school achievement, amongst others because the help fathers provide and the time that fathers practice with their children can improve their children's skills [31]. In addition, the greater involvement of fathers in schoolwork could positively impact children's educational achievements, because greater school involvement of parents is associated with more ambitious school aspirations, greater motivation to perform well in school, and higher school attendance [32]. The rationale behind these findings is that greater involvement of parents is indicative of the value they attach to educational achievements, and children internalize these values [31]. Previous research found that children who have fathers that are more involved in their schoolwork obtained higher grades in high school [21, 29, 30] (correlation of 0.113 between father involvement and academic achievement [29] These positive associations remain after accounting for the involvement of mothers [21, 30]. Besides fathers' involvement in school (work), the frequency of joint activities undertaken by father and child is also associated with children's educational attainment [33, 34]. Previous studies show that children who spend more time alone with their father score higher on cognitive tests [33] (also when controlling for the involvement of the mother). In addition, studies show that the more time fathers spend with their children on care-related tasks as well as playing and reading, the better children fare in school tasks [34]. Thirdly, there is empirical evidence showing that fathers' educational attainment is related to fathers' involvement, as fathers who finished more than 16 years of education spend 4,76 more hours on childcare than high school dropouts [18]. Highly educated fathers are generally more involved in their children's lives, amongst others because these fathers feel more confident in helping their children with schoolwork [35, 36], because they have more intensive parenting ideologies [5] and they have more time available [18].

Based on the abovementioned theoretical and empirical work, it is likely that fathers' involvement mediates the association between fathers' and children's educational attainment. Our first hypothesis therefore reads: *Fathers' involvement (school-specific involvement and leisure involvement) is an underlying mechanism for the association between fathers' educational attainment and children's educational attainment (H1).* A small number of studies have shown that fathers' involvement indeed serves as a mediator between educational attainment

and child outcomes [29, 37–39]. A clear limitation of these studies is that they did not control for children's genetic characteristics, and therefore might not have been able to obtain a reliable understanding of the role father involvement plays in the intergenerational transmission of educational attainment.

## Genetic influences as mechanism underlying the intergenerational transmission of educational attainment

A large number of twin studies showed that educational attainment is approximately 40% heritable, that is, 40% of the variation in education is accounted for by genetic variation [6]. This heritability can be explained partly by the heritability of intelligence, but also by the heritable component of amongst others personality traits, self-efficacy, and behavioural problems [40]. Children with genes that are positively related to higher educational attainment tend to be more open, agreeable, conscientious, and show more academic motivation (correlations between 0.01 and 0.03), which are linked to better educational achievements [41]. As parents and children are 50% genetically similar, part of the intergenerational transmission of education is posed to be through genetic influences. Several previous studies showed that genes can account for part of the intergenerational transmission of education [42–44]. Based on the above-described literature we expect to find that *genetic influences are an underlying mechanism for the association between fathers' educational attainment and children's educational attainment (H2)*.

## Three types of correlations between genetic influences and father involvement

There are three different reasons why we can expect to find a correlation between father involvement and children's education PGS. These three reasons are related to the three types of rGE that can theoretically be distinguished [45]. The first type is an *evocative gene-environment correlation*: parents will behave differently to a child based on the child's characteristics (which are genetically influenced) [46]. In our case, children's education PGS does not only capture the intelligence of the child but also personality traits, academic motivation, behavioural problems, and self-efficacy [40, 41]. It is therefore likely that children with a higher education PGS more often evoke their father's involvement (in particular with school), as these children are also more likely to be more interested in learning. If this is true, this would result in a positive rGE.

The second type is an *active gene-environment correlation*: children with higher education PGS might not only be more likely to be more interested in learning, but they might also be more prone to actively seek help from their parents with homework or discuss school matters with their father, which may then result in greater involvement of the father.

The third type is a *passive gene-environment correlation*: children inherit half of their genes from each biological parent, and, if children live with their biological parents, these same parents also rear them and shape their environment. This can result in a correlation between the child's education PGS and father's involvement in two ways. Firstly, children with a high education PGS are more likely to have a parent with a high education PGS, and the parent's education PGS is not only associated with their own educational attainment but also their parenting practices [9]. Secondly, children with a higher education PGS generally have a highly educated father, and because of reasons such as status maintenance motives [47], these highly educated fathers are more likely to be more involved in their children's lives.

Based on the abovementioned theoretical considerations, *we expect that father involvement (school-specific involvement and leisure involvement) is positively correlated with children's*

*education PGS in their relation to education (H3).* Several previous studies showed a correlation between genes and different aspects of the family environment, such as parental sensitivity, warmth, stimulating parenting, and parental SES [7–9, 48], and one study looked specifically at the correlated effect of genes and family environment on education [10]. No study has looked at the correlation between genes and father involvement in their relation to education.

## Implications of correlations between genetic factors and father involvement for understanding the mechanisms underlying the intergenerational transmission of educational attainment: Genetic confounding

In the context of the expected correlations between genetic factors and father involvement, it might be the case that part of the effect of father involvement is driven by genetic factors, so-called genetic confounding; both fathers' involvement as well children's educational attainment are accounted for by the same genetic factors (see Fig 1 middle part for a graphical representation). This is due to the passive rGE. For example, the same genes that are associated with higher education, are also associated with greater involvement of fathers through personality traits or a sense of responsibility of the father. Previous research by Wertz and colleagues showed that controlling for genetics reduced the association between parental warmth/sensitivity and child's educational achievement by about 8% [8]. In line with this finding, we expect that *the genetic influences partly account for the behavioural mechanism underlying the intergenerational transmission of educational attainment (H4).*

## Implications of correlations between genetic factors and father involvement for understanding the mechanisms underlying the intergenerational transmission of educational attainment: Social confounding

Alternatively, it might be the case that part of the association between children's education PGS and children's educational attainment is confounded by the environment, in our case father involvement (see Fig 1 bottom part for a graphical presentation of this hypothesis). The rationale for the existence of social confounding is that the education PGS is based on the GWAS of a large number of SNPs that are associated with educational attainment. Yet, these associations do not imply causation, and the pathways from genetic variants to education are diverse [49]. GWAS studies that are used to create PGSs cannot distinguish between, on the one hand, associations between genes and education through personal traits, such as intelligence and motivation, and on the other hand, associations between genes and education due to the environment, such as the family environment and parenting practices. As such, part of the effect of our genetic factors might be driven by our behavioural factors; children's education PGS is associated with children's educational attainment, because children's education PGS is associated with the parenting of the child's father, which is driven by his education PGS, and it is father involvement that is shaping children's educational attainment. Previous research found that the association between mother's genes and children's education is partly accounted for by mothers' cognitively stimulating parenting [8] and that the association between children's own genes and their educational attainment is reduced once family SES and household chaos are taken into account [10]. Studies also showed that the genes that parents do not transmit to their children are associated with their children's educational attainment, which is likely due to the environment parents provide to their children [50–52]. This hints towards the idea that also the genes that parents do transmit to their children partly

influence their children through the home environment. Based on the above, we expect that *father involvement partly accounts for the genetic mechanism underlying the intergenerational transmission of education (H5)*.

## Zooming in: Understanding which type of gene-environment correlation is at play

Previous research has not always been able to differentiate passive rGE from active and evocative correlations, which is unfortunate if one wants to understand why this correlation exists. In our study, we can differentiate active/evocative from passive rGE, by using a subsample of our data that includes sibling data. The presence of within-family correlations between children's education PGS and father involvement (i.e. differences in the education PGS of siblings correlate with differences in father involvement towards these siblings) would imply active/evocative rGE. However, if within families we find a substantially smaller correlation between father involvement and children's genes as compared to between-families, this would indicate that the correlation is mainly driven by characteristics of the family, which is indicative of passive rGE. Although no previous studies examined passive versus active rGE regarding father involvement, research found support for child evoked rGE, showing that children's genotype evokes parental warmth, sensitivity, harsh discipline, negative effect [48], but also passive rGE has been found for example regarding warmth, sensitive and stimulating parenting [8, 9, 53]. We expect to find *passive rGE for both indicators of father involvement (father's school-specific involvement and leisure involvement) (H6)*. In addition, given that father involvement might be explicitly activated or evoked based on children's genetic disposition towards school, we expect that *child-evoked/active rGE will only be found for father's school-specific involvement (H7)*.

## Current study

This study aims to gain a better understanding of the mechanisms underlying the intergenerational transmission of educational attainment by scrutinizing the interrelatedness of genetic influences and father involvement. We will answer 5 related research questions: 1) To what extent do genes and father involvement independently account for the intergenerational transmission of education, 2) to what extent do genes and behaviour correlate in their relation to educational attainment, 3) to what extent do genes account for part of the behaviour mechanism ('genetic confounding'), 4) to what extent does behaviour account for part of the genes mechanism ('social confounding') and 5) to what extent can we distinguish between passive and active/evocative rGE? We will examine these research questions using data from the National Longitudinal Study of Adolescent to Adult Health (Add Health) of 5,021 genotyped individuals. These data contain information on father involvement, as well as the same information for mothers. Controlling for maternal involvement is important, as this allows us to isolate the independent effects of paternal involvement. Due to amongst others behavioural contagion, in which parents copy each other's behaviour, mothers and fathers within the same household are more likely to show similar behaviours [54, 55] also concerning parental involvement [56]. Because our sample of respondents has been genotyped, we can include the child's education PGS, which captures the additive effect of common genetic influences on education. In addition, using sibling data, available for a subsample within our data, we can test the hypothesized causes of any correlation between father involvement and education PGS. These sibling data allow us to unravel whether observed correlations between father involvement and education PGS are attributable to passive versus active/evocative gene-environment correlations, which have not been assessed yet for father involvement. Respondents in our sample are followed from age 12–19 up to age 32–42.

## Methods

### Data

The Add Health data is a longitudinal study of adolescents from the United States who were in grades 7 to 12 in wave 1 in 1994/95 [57]. The data were collected using a school-based stratified sample, in which 80 high schools in the United States were selected, and from these 80 high schools, 20,745 adolescents in grades 7 to 12 participated [57]. In this first wave, respondents were between 12 and 19 years old, with the majority between 14 and 18 years old. In the second wave, in 1996, respondents were between 13 and 21 years old. In the third wave, in 2001/02, respondents were between 18 and 26. At wave 4 in 2008/09, respondents were between 24 and 32. Finally, during wave 5 in 2016/18, respondents were between 32 and 42 years old.

From the 20.745 respondents, we selected respondents who were followed up to wave 4 or 5, since these respondents are between the ages 24 and 42 and were thus most likely to have finished their educational training. This restriction reduced our sample size to 17,535. After this, we selected only respondents who reported the educational attainment of their biological father and biological mother, which narrowed down our sample size to 14,712. After this, we only included respondents with an education PGS, which reduced the sample to 5,021 individuals. Finally, we removed individuals with missing information on parental involvement, resulting in a final sample size of 4,579 respondents. Although Add Health is a nationally representative sample, selecting only respondents with information about their own educational level, the educational level of their parents, and information about parental involvement resulted in a sample that was relatively higher educated and contained respondents who relatively more often lived with both their parents during the first wave of data collection. Furthermore, selecting only genotyped respondents led to a sample that only contained White respondents.

In total 96% of the participants in wave 5 gave saliva, and 80% of them consented with long-term archiving of their data and were eligible for genome-wide genotyping [58]. This resulted in approximately 12,200 genotyped respondents. After quality control, data were available for 9,974 individuals (please see Highland et al [59] for details on quality control). Genetically-determined non-European descent individuals were removed from the sample, as PGSs based on a GWAS with European-descent individuals is much less suitable for non-European descent individuals, leaving 5,787 individuals. After additional quality control, 5,690 individuals remained with valid genetic information (of which we use 4,579 who also have information of their educational level and parental involvement available).

Individuals were genotyped using the Illumina Omni1-Quad BeadChip (80% of the sample) and the Illumina Omni2.5-Quad BeadChip (20% of the sample). Before quality control, 609,130 SNPs that were common across both genotyping platforms were available. After quality control (call rate $<0.98$, HW p-value $<10^{-4}$, MAF$<0.01$) 346,754 SNPs remained and were used for imputation. Imputation was done using the Haplotype Reference Consortium (HRC) v1.1 European reference panel.

Sibling data: The Add Health contains in total 3,139 sibling pairs, consisting of full-siblings, half-siblings, and MZ and DZ twin pairs and unrelated siblings. Reducing this sample to only sibling pairs with education PGS available reduces the sample size to 619 sibling pairs, of which 429 full siblings and DZ twin pairs, which is further reduced to 380 sibling pairs once we only include those who have information of father involvement available.

### Ethical considerations

The Medical Ethics Review Board of the Erasmus Medical Center Rotterdam considered whether or not this research falls within the scope of the Medical Research Involving Human

Subjects Act (WMO). It was concluded that the research is not a clinical research with test subjects as meant in the Medical Research Involving Human Subjects Act (WMO). Therefore, the Medical Ethics Review Board of the Erasmus Medical Center Rotterdam had no task in reviewing the protocol. Therefore it was concluded that we were allowed to conduct the research. Add Health participants provided written informed consent for participation in all aspects of Add Health.

## Measures

Years of education of the child: In waves 4 and 5 respondents were asked about the highest level of education that they completed. Answer categories ranged from '8th grade or less', to 'completing a post-baccalaureate professional degree'. We recoded this variable to years of education. Response options and the years of education that correspond to them (in parentheses) were: 8th grade or less (8), some high school (10), high school graduate (12), some vocational/technical training (13), completed vocational/ technical training (14), some college (14), completed college (16), some graduate school (17), completed a master's degree (18), some graduate training beyond a master's degree (19), completed a doctoral degree (20), some post-baccalaureate professional education (18), and completed post-baccalaureate professional education (19). This is in line with previous studies that coded years of education in AddHealth [26, 60, 61]. To reduce missing data, for those respondents who did not give information in wave 5, we used the information from wave 4.

Years of education of parent: In wave 1, respondents were asked about the educational attainment of their biological parents. Answer categories ranged from '8th grade or less', to 'professional training beyond a four-year college or university'. We recoded this variable to years of education. Response options and the years of education that correspond to them (in parentheses) were: never went to school (0), 8th grade or less (8), more than eighth grade, but did not graduate from high school (10), went to a business, trade, or vocational school instead of high school (10), high school graduate (12), completed a GED (12), went to a business, trade, or vocational school after high school (14), went to college, but did not graduate (14), graduated from a college or university (16), professional training beyond a four-year college or university (18). For a subset of the respondents, parents provided information on their educational attainment themselves. Our robustness checks show that the correspondence between child reports and parent reports is high: correlation of 0.87 for mothers and 0.79 for fathers, see Supplementary Material part 3 in S1 File).

Father's school-specific involvement: In wave 1, respondents were asked if, in the past four weeks, they talked with their father about schoolwork or grades (yes or no), worked with their father on a project for school (yes or no), and talked with their father about other things they are doing in school (yes or no). These measures were added up, ranging from 0 for children who did none of these school-related activities with their father to 3 for those who did all these activities. The Cronbach's alpha for this scale is 0.77.

Father's leisure involvement: In wave 1, respondents were asked if in the past four weeks they went shopping with their father (yes or no), played sports with their father (yes or no), talked with their father about someone they're dating or a party they went to (yes or no), went to a movie, play, museum, concert or sports event with their father (yes or no) or talked about a personal problem with their father (yes or no). These activities were added up, ranging from 0 if the respondent did none of these activities with their father, to 5, if they did all these activities with their father. The Cronbach's alpha for this scale is 0.78.

Polygenic score (PGS) for years of education: We include PGS for years of education based on the GWAS conducted among 1.1 million individuals [26]. The PGS was constructed by the

Social Science Genetic Association Consortium (SSGAS), based on the discovery sample that did not include Add Health [58] and is provided by Add Health. This PGS was created using LDpred, which uses all SNPs and weights them according to their conditional effect, given all other SNPs.

**Controls.** Father residence: We distinguish between whether the child lived with the biological father at wave 1 (resident father = 1) or not (non-resident father = 0).

Principal components: To control for population stratification, which is the case if certain SNPs are more common in certain ancestry groups than others, we control for the first 10 genetic principal components (PCs). These genetic PCs were created by the SSGAS [58].

In our models, we control for the *age the respondent had at the first interview*, which ranged between 12 and 21, with the majority of the respondents between 14 and 18. We control for the *sex of the respondent*, and whether or not the respondent was *enrolled in school* at the last wave of data collection.

To examine the unique contribution of father involvement, we control for certain characteristics of the mother, namely *years of education of the mother* and maternal involvement, namely mother's school-specific involvement, mother's leisure involvement, and whether or not the child lived with the biological mother at wave 1.

## Analyses

**Path model to test for mediation (hypothesis 1 and 2) and confounding (hypothesis 4 and 5).** To estimate the extent to which fathers' involvement and children's PGS for years of education mediated the relationship between fathers' years of education and children's years of education (hypothesis 1 and 2), a path model was estimated using the Lavaan package in R [62]. Since the respondents in our sample are not independent but nested within households, we ran a multilevel path model of individuals nested within households [63].

In our path model, we simultaneously tested the direct effect of the father's years of education on the years of education of his child, as well as the mediated effect via father's involvement and the child's PGS for years of education. These mediated effects (hypothesis 1 and 2) are estimated by multiplying the coefficient of (a) the independent variable on the mediators and (b) the mediators on the outcome [64].

Because we wanted to assess potential confounding of genetic effects by father involvement and vice versa (hypothesis 4 and 5), three nested path models were fitted, first a multiple mediation model in which only the two aspects of father involvement are assessed simultaneously, second a mediation model in which the role of children's PGS for years of education is examined, and third a multiple mediation model that includes both father involvement and children's education PGS. To quantify the extent to which the effect of father involvement in explaining the intergenerational transmission of education is confounded by the education PGS (hypothesis 4), we compare the coefficients of father involvement between the first model (in which only father involvement is included as a mediator) and the third model (in which also the education PGS is included) [64]. The other way around, to quantify the extent to which the effect of the education PGS is partly socially confounded (hypothesis 5), and can be accounted for by father involvement, we compare the coefficient of the education PGS between the second model (in which only the education PGS is included as a confounder), and the third model (in which both the education PGS and father involvement are included). We regress all our control variables on educational attainment of the child, and we include the first 10 genetic PC's on the path from father's education to the child's PGS.

**rGE (hypothesis 3, 6, and 7).** Correlated effects: To examine to what extent the education PGS and father involvement correlate in their relation to educational attainment, we estimated

the correlation between children's years of education predicted from the PGS model and children's years of education predicted from the father involvement model. To this end, we first regressed children's years of education on both parents' years of education, the first 10 genetic PCs, and all other control variables using a multilevel model that takes into account the nested structure of the data. The individual level residuals from this model were used and regressed on the education PGS (model 1), father's school-specific involvement (model 2), and father's leisure involvement (model 3). Finally, we assessed the correlation between the predicted values from model 1 with model 2 and model 3, the correlated effects. This approach allows us to not only assess the correlation between the PGS and father involvement, but also whether and to what extent their associations with years of education correlate.

To examine active/evocative and passive rGE, we estimate rGE between and within families. Between families, we cannot simply examine the correlation between father involvement and the education PGS, as we not only have to control for spurious associations based on ancestral differences but also because we have to take the nested structure of the data into account. Therefore, we estimated multilevel regression models in which we explain the two measures of father involvement by the education PGS while controlling for the first 10 genetic PCs (hypothesis 3).

To assess the rGE within families, we do not have to control for the first 10 genetic PCs. The reason that we do not have to control for the first 10 genetic PC's is that siblings share their ancestry, and the genetic PCs are used to control for ancestral differences. We also do not have to take into account the nested structure of the data as there is only one sibling pair per family. Therefore, we estimated linear regression models in which we explain the difference in father involvement between siblings by the difference in the education PGS between siblings. To distinguish between active and passive rGE (hypotheses 6 and 7), we will compare estimates of rGE within and between families.

## Results

### Univariate descriptives

Sample descriptives can be found in Table 1. The respondents in our sample finished 14.63 years of education on average, which equals to some years of education after finishing high

**Table 1. Univariate descriptives of the sample.**

|                                        | Mean  | SD   | Min   | Max   |
|----------------------------------------|-------|------|-------|-------|
| Years of education child               | 14.62 | 2.32 | 8     | 20    |
| Years of education father              | 13.52 | 2.54 | 0     | 18    |
| Years of education mother              | 13.48 | 2.35 | 0     | 18    |
| PGS education child                    | 0.55  | 0.15 | 0     | 1.05  |
| Father's school-specific involvement   | 1.16  | 1.01 | 0     | 3     |
| Father's leisure involvement           | 1.44  | 1.29 | 0     | 5     |
| Mother's school-specific involvement   | 1.32  | 1    | 0     | 3     |
| Mother's leisure involvement           | 2.08  | 1.21 | 0     | 5     |
| Age first interview                    | 16.01 | 1.7  | 12    | 21.33 |
| Age asked education                    | 35.83 | 4.13 | 25    | 43    |
|                                        | Yes n | %    | No n  | %     |
| Enrolled in educ last wave             | 355   | 7.75 | 4223  | 92.25 |
| Live with father at the first wave     | 3427  | 74.84| 1152  | 25.16 |
| Live with mother at the first wave     | 4181  | 91.31| 398   | 8.69  |

school (12 years equals to finishing high school and 16 years equals to finishing a bachelor's degree). Their parents on average were in school a bit shorter (on average 13.5 years). Fathers were on average less involved with their children's school and leisure activities than mothers were. We expected to find greater variation in father involvement compared to mother involvement. In contrast, however, the variation in these two aspects of parental involvement did not substantially differ between fathers and mothers. Children's age at the first interview was, on average, 16 years and they were almost, on average, 36 at the final wave when they were asked about their final education. 71% lived with their father at the first wave, and 29% did not, while a little under 10% did not live with their mother at the time of the first wave of data collection.

## Bivariate descriptives

Fig 2 and S2 Table show the correlation between the variables in our analyses. Children's years of education is positively correlated with the education of the father, mother and with children's education PGS (0.45, 0.43 and 0.38 respectively). Furthermore, both father's and child's years of education are positively correlated with father's school-specific involvement (0.14 and 0.17 respectively) and father's leisure involvement (0.14 and 0.12 respectively). The child's

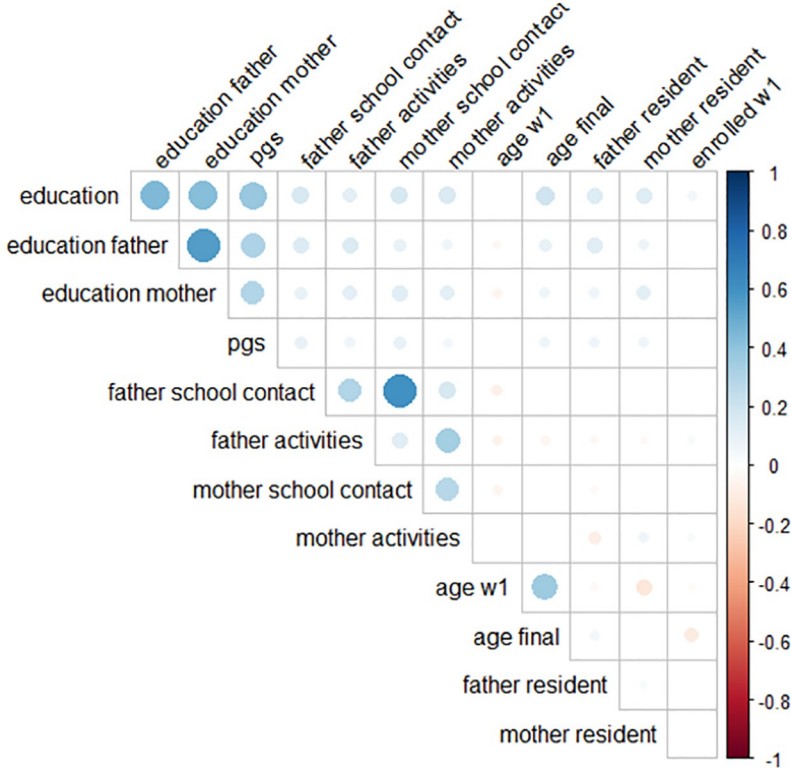

**Fig 2. Correlation table graphically displayed.** Non-significant correlations are not displayed. The size and color represent respectively the strength and direction of the correlation. Education = Years of education child; Education father = Years of education father; Education mother = Years of education mother; pgs = Child's Polygenic score for education; father school contact = father's school-specific involvement; father activities = father's leisure involvement; mother school contact = Mother's school-specific involvement; mother activities = mother's leisure involvement; age w1 = age at wave 1; age final = age at the final wave; father resident = lived with father at wave 1, 0 = no 1 = yes; mother resident = lived with mother at wave 1, 0 = no 1 = yes; enrolled w1: was the respondent enrolled in education at wave 1, 0 = no, 1 = yes.

education PGS is positively correlated with father's school-specific involvement (0.09) and father's leisure involvement (0.06). There is also a positive correlation between the two different aspects of father involvement of 0.30 (father's school-specific involvement and father's leisure involvement).

## Part 1) Regression results: Mediation

Our first two hypotheses focused on the extent to which the association between fathers' years of education and children's years of education is mediated by father involvement (hypothesis 1) and the child's education PGS (hypothesis 2). To this end, we estimated path models in which we examined the direct association between father's educational attainment and children's educational attainment and the significance of the indirect effects via father involvement and education PGS. All measures are standardized. The results are displayed in S1 Table and Fig 3. The total effect of father's education on child's education is 0.303 (Se 0.016). Father's school-specific involvement mediates 2.3% of the intergenerational transmission (0.007 of the total effect of 0.303), father's leisure involvement mediates 1.3% (0.004 of 0.303), and the education PGS mediates 21.45% (0.065 of 0.303). These findings support both hypotheses 1 and 2 and show that both genes and father involvement are significant mediators. In addition, our findings show that the education PGS mediates a much larger proportion of the intergenerational transmission of father's years of education than our two measures of father involvement.

## Part 2) rGE

Table 2 displays the correlations between the child's education PGS and our two dimensions of father involvement. The correlated effects are displayed in the left panel of Table 2. We find correlated effects between father's school-specific involvement and the education PGS of 0.092, and somewhat smaller positive correlated effects between father's leisure involvement and the education PGS of 0.063 (see Table 2 left panel).

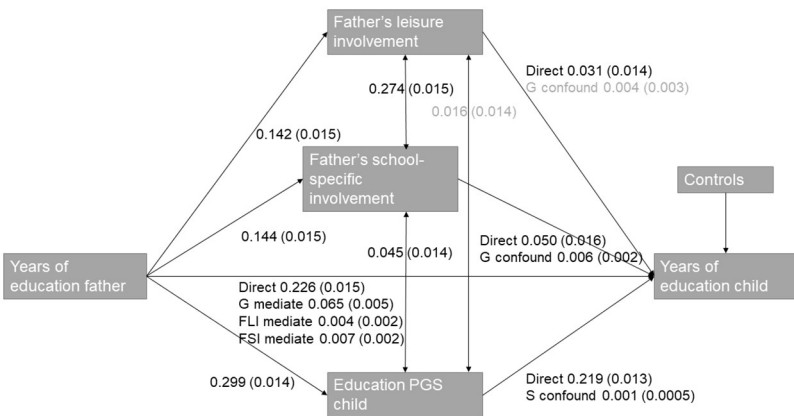

**Fig 3. Graphical presentation of the mediation and confounding results.** G mediate refers to the part of the effect that is genetically mediated, FLI mediate refers to the mediated effect by father's leisure involvement, FSI mediate refers to the mediated effect by father's school-specific involvement, G confound to the genetically confounded part and S confound to the social confounding by father involvement. The controls included in this model are years of education mother, age child at wave 1, child's sex, resident father or not, enrolled in education, resident mother or not, mother's school involvement, mother's leisure involvement and the first 10 genetic PC's. All measures are standardized. Standard errors are displayed in parentheses, and non-significant estimates in light grey.

**Table 2. Correlations between education PGS and father involvement: Correlated effects, between-family, and within-family correlations.**

| | Education PGS | | | | | |
| | Correlated effects | | Between families | | Within families | |
| | Corr. | 95% CI | β | 95% CI | Corr. | 95% CI |
|---|---|---|---|---|---|---|
| Father's school-specific involvement | 0.092*** | 0.063–0.121 | 0.088*** | 0.059–0.118 | 0.096 | -0.004–0.194 |
| Father's leisure involvement | 0.063*** | 0.034–0.092 | 0.058*** | 0.029–0.088 | 0.006 | -0.094–0.107 |
| N individuals | 4579 | | 4579 | | 760 | |
| N families | 4154 | | 4154 | | 380 | |

*p<0.05,

**p<0.01,

***p<0.001.

The between-family associations are based on multilevel regression models, in which father involvement is explained by the education PGS while controlling for the first 10 genetic PCs and taking into account the nested structure of the data. We display the standardized beta coefficients. The within-family estimates refer to the correlation between the difference in education PGS between siblings with the difference in father involvement between siblings.

## Part 3) Genetic confounding

Thirdly, we were interested in genetic confounding; the extent to which accounting for genetic influences reduces the association between father involvement and children's educational attainment. Results are displayed in Fig 3. The association between father's school-specific involvement and children's educational attainment is reduced from 0.056 to 0.050 when child's education PGS is added to the model, which implies 10.7% genetic confounding, while the association between father's leisure involvement and children's educational attainment is not significantly reduced when including children's education PGS. The former shows that a small, but nonnegligible, part of the role that father's school-specific involvement plays in the intergenerational transmission of years of education is genetically confounded, which supports hypothesis 4. The latter shows that the role that father's leisure involvement plays in the intergenerational transmission of years of education is not genetically confounded, which contrasts with hypothesis 4.

## Part 4) Social confounding

Furthermore, we were interested in social confounding: the extent to which the role of children's genes in the intergenerational transmission of years of education is reduced when information on father involvement is included in the model. Results are again displayed in Fig 3. Our results showed that, once father involvement is added to the model, the role of children's education PGS as the underlying mechanism is only slightly reduced, from 0.220 to 0.219, which implies a mere 0.5% social confounding. Furthermore, this social confounding only holds for the father's school-specific involvement. These findings support hypothesis 5 but are very small in magnitude.

## Part 5) rGE within versus between families

Comparing the between-families and within-families associations between father involvement and children's education PGS provides insights into the extent to which the rGE is either active/ evocative or passive. For father's school-specific involvement, we do not find evidence for within-family correlation, but we do find a between-family correlation (0.088, 95% CI 0.059–0.118), see Table 2. Similarly, for father's leisure involvement, we only find a between-family correlation (0.058, 95% CI 0.029–0.088), implying passive rGE from the side of the child. These

findings are in line with hypothesis 6, showing passive rGE, but are in contrast to hypothesis 7 on active or child-evoked rGE for father's school-specific involvement.

## Robustness checks

We conducted several additional analyses to assess the robustness of our findings. To assess the robustness of our findings with respect to the education PGS, we tested whether and to what extent our findings were inflated by indirect effects, due to population stratification, or due to assortative mating. We found that the within-family effect of the education PGS on the education of the child was smaller than the between family effect, yet it remained significant, which shows that controlling for indirect effects and population stratification does not fully explain our findings regarding the association between the education PGS and years of education. Our findings indicate that assortative mating could to a small extent result in an overestimation of the effect of the education PGS (see S1 Robustness checks in S1 File). We furthermore assessed whether the association between our measures of father involvement and education depends on the age of the respondents during the first interview. Our robustness checks reveal it does not (see S2 Robustness checks in S1 File). We also assessed whether multi-collinearity issues arise when father involvement and mother involvement were added simulta-neously to the model. Our analyses revealed no multicollinearity issues (see S2 Robustness checks in S1 File). We furthermore assessed whether the results in our sample with only resi-dent fathers differed from our findings in the complete sample and found that the results are largely comparable and do not change any substantive conclusions. Finally, we examined the robustness of our findings by assessing whether our results differ when we use a parent-report or child-report on parental education. Our analyses revealed that our conclusions are similar, irrespective of the reported used (see S3 Robustness checks in S1 File).

## Conclusion and discussion

This study aimed to gain a better understanding of the mechanisms, and their interrelatedness, underlying the intergenerational transmission of educational attainment. To this end, we answered five interrelated questions.

Our first question was to examine whether and to what extent father involvement and children's education PGS were unique and independent mechanisms underlying this intergen-erational transmission of educational attainment. Our results revealed that both dimensions of father involvement as well as children's education PGS were unique mechanisms. Noteworthy is that our results indicated that children's education PGS was a much more important under-lying mechanism than the two dimensions of father involvement we considered in the current study. Our relatively small effects of the family environment compared to the genetic compo-nent is in contrast to findings from Allegrini and colleagues [10], who found that multiple PGSs can account for 15% of the variance in educational attainment, while environmental factors can account for 28%. The higher percentage for environmental factors found in their study is likely attributable to the fact that they used a much broader indication of the family environment, consisting of amongst others the educational attainment of both parents, employment status of parents, chaos at home, and important life events.

Our second question focused on the extent to which children's education PGS and father involvement correlate as mechanisms underlying the intergenerational transmission of educational attainment. Our results indicated that both mechanisms indeed correlate. These findings add to the previous research that shows a correlation between other aspects of parent-ing and the education PGS, such as parental warmth and sensitivity [9]. This correlation implies that many children are either growing up with double disadvantages–having both a

less involved father and a lower education PGS–or double advantages–a highly involved father and a higher education PGS. This highlights the importance for social scientists to take genetic influences into account when examining the importance of parenting in the intergenerational transmission of education, and for genetic scientists to take into account social pathways.

Our third question centred on genetic confounding; to what extent does children's education PGS account for the role that father involvement plays in the intergenerational transmission of educational attainment. Our results indicated that genetic confounding plays a part, albeit small, in accounting for the role that fathers' school-specific involvement plays as an underlying mechanism in the intergenerational transmission of educational attainment. This indicates that findings from the field of behavioural sciences have likely, at least to some extent, overestimated the role that fathers' school-specific involvement plays in the intergenerational transmission of educational attainment. It furthermore suggests that part of "what we think of as measures of 'environment' are better described as external factors that might be partly under genetic control" [65]. However, and intriguingly, the role of fathers' leisure involvement as an underlying mechanism hardly changed when information on children's genes was added to the model. Furthermore, fathers' school-specific involvement was not fully mediated by the children's education PGS, indicating that also school-specific involvement independently explains the intergenerational transmission of education. Also, our suggestive finding of active gene-environment correlation within our sibling sample does not exclude the possibility that genetic confounding is largely caused by child evoked genetic correlation. Our findings regarding the relevance of father involvement in the intergenerational transmission of educational attainment therefore suggest that father involvement should not be dismissed as a potential candidate for an intervention to aid in breaking the intergenerational cycle of (dis) advantage. Several interventions, such as educational sessions on parenting skills, as well as policies, such as parental leave reserved for fathers, have been proven to increase father's involvement [66, 67].

Our fourth question revolved around behavioural confounding; to what extent is the role that children's education PGS plays in the intergenerational transmission of educational attainment accounted for by father involvement. Our results indicate that behavioural confounding plays a negligible role. These findings differ from the findings from previous research that examined social confounding of genetic effects [8, 10]. Again, these differences are likely attributable to differences in how broadly the family environment was defined.

Fifth, we assessed active/evocative versus passive rGE by looking at correlations within and between families. Our findings reveal that the correlation between the education PGS and activities between father and child is more passive from the perspective of the child, given that we do not find a within-family correlation. This indicates that children with a high education PGS grow up in families with highly educated fathers/fathers with a high education PGS, who have a more active parenting style and therefore perform more activities with their child.

Some limitations need to be considered when interpreting these findings. First, we believe that our measures of father involvement are relevant and cover important dimensions of involvement of fathers, yet there are other aspects of father involvement that are also relevant that we did not cover. Our measures largely tap into the quantity of involvement (how much do fathers and children discuss school matters and undertake activities together). Other studies have emphasized that pertaining to children's educational outcomes, the quality of parental involvement is also important [68]. It is therefore likely that the current study only yielded an underestimation of the role father involvement plays in the intergenerational transmission of educational attainment.

Similarly, our genetic measure only captures a part of the genetic component of education, as twin studies showed a heritability of 40% [6] while the education PGS accounts for approximately

10%. This discrepancy between relatively high heritability estimates from twin studies and lower explained variance from PGSs has been called 'missing heritability' and has been found for many traits [69]. Possible reasons are amongst others that PGSs only captures common genetic variants and only additive effects, while rare variants and non-additive effects might be relevant as well. Therefore, our estimates of genetic mediation of the intergenerational transmission of education, and genetic confounding of father involvement, are underestimations and only capture part of the genetic effect. Furthermore, it is important to take into account that PGSs do not imply genetic causality, but are merely based on associations between genetic variants and phenotypes (in our case educational attainment). Therefore, we cannot say that the mediating role of the PGS indicates that this part of intergenerational transmission is due to genes. In fact, this study shows that one of the pathways between PGS and educational attainment is environmental, namely through father involvement.

A suggestion for future research would be to include the genotype of the father, as is done with mother's genotypes in the study by Wertz and colleagues [8]. By incorporating father's genotype, on would be able to examine to what extent the relationship between father's education and father's involvement is due to genetic transmission. It would furthermore allow one to examine the extent to which the non-transmitted part of the father's genotype is related to the child's education, and to what extent the association between the non-transmitted part of the genotype and the child's education is accounted for by parental involvement.

When interpreting our findings regarding father involvement, we must consider that our data allow us to investigate the role of fathers in the United States in 1994. At that time, involvement of fathers was much lower than in more recent cohorts [70]. The greater involvement of fathers in more recent cohorts could imply that fathers play an even bigger role in the educational achievements of their children, and therefore in the intergenerational transmission of education in these cohorts. However, the group of fathers that exhibited strong involvement in their children's lives was most likely more selective in older cohorts than in recent ones. Consequently, it might also be the case that father involvement played a less substantial role as underlying mechanism in more recent cohorts than in older ones.

Our study is limited by the fact that our sample only includes respondents of European ancestry (i.e. White respondents). The reason is that the GWAS for educational attainment is based only on European ancestry individuals, and PGSs created from such GWASs have lower predictive power among non-European samples [71]. Therefore, we cannot generalize our findings to other ancestry groups. For this reason, we also cannot use our findings to explain differences in educational achievement between different ancestry groups.

To summarize, we find that both genes and father involvement are underlying mechanisms in the intergenerational transmission of educational attainment, that these mechanisms are correlated with each other, and that part of the role that fathers' school-specific involvement plays as underlying mechanism is confounded by children's education PGS. Our findings underscore the need to control for genetic effects in studies that examine the role of parenting in the intergenerational transmission of inequality, but also the need to control for parental involvement and the family environment in general when considering the role that genes play in this intergenerational transmission. Thus, our study underscores that in order to fully understand the mechanisms that underly intergenerational reproduction of (dis)advantages, scholars need to integrate both insights and data from different disciplines.

## Supporting information

**S1 File.**
(DOCX)

## Acknowledgments

Add Health was designed by J. Richard Udry, Peter S. Bearman, and Kathleen Mullan Harris at the University of North Carolina at Chapel Hill.

## Author Contributions

**Conceptualization:** Renske Marianne Verweij, Renske Keizer.

**Formal analysis:** Renske Marianne Verweij.

**Funding acquisition:** Renske Keizer.

**Investigation:** Renske Marianne Verweij, Renske Keizer.

**Methodology:** Renske Marianne Verweij, Renske Keizer.

**Validation:** Renske Marianne Verweij.

**Writing – original draft:** Renske Marianne Verweij, Renske Keizer.

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
