## [Decision Letter · Decision Letter 0]

17 Jun 2022

PONE-D-22-08657The intergenerational transmission of educational attainment: A closer look at the (interrelated) roles of paternal involvement and genetic inheritancePLOS ONE

Dear Dr. Verweij,

Thank you for submitting your manuscript to PLOS ONE. After careful consideration, we feel that it has merit but does not fully meet PLOS ONE’s publication criteria as it currently stands. Therefore, we invite you to submit a revised version of the manuscript that addresses the points raised during the review process.

We look forward to receiving your revised manuscript.

Kind regards,

Marie-Pierre Dubé, PhD

Academic Editor

PLOS ONE

Journal Requirements:

"The present study was supported by a grant from the Netherlands Organization for Scientific Research to RK (NWO MaGW VIDI; grant no. 452-17-005) and by a grant from the European Research Council to RK (ERC StG; grant no. 757210). Both authors declare no conflict of interest. 

Add Health is directed by Robert A. Hummer and funded by the National Institute on Aging cooperative agreements U01 AG071448 (Hummer) and U01AG071450 (Aiello and Hummer) at the University of North Carolina at Chapel Hill. Waves I-V data are from the Add Health Program Project, grant P01 HD31921 (Harris) from Eunice Kennedy Shriver National Institute of Child Health and Human Development (NICHD), with cooperative funding from 23 other federal agencies and foundations. Add Health was designed by J. Richard Udry, Peter S. Bearman, and Kathleen Mullan Harris at the University of North Carolina at Chapel Hill."

"The present study was supported by a grant from the Netherlands Organization for Scientific Research to RK (NWO MaGW VIDI; grant no. 452-17-005) (https://www.nwo.nl/en) and by a grant from the European Research Council to RK (ERC StG; grant no. 757210) (https://erc.europa.eu/)."

Reviewers' comments:

Reviewer's Responses to Questions

**Comments to the Author**

1. Is the manuscript technically sound, and do the data support the conclusions?

Reviewer #1: Yes

Reviewer #2: Partly

Reviewer #3: Yes

Reviewer #4: Yes

2. Has the statistical analysis been performed appropriately and rigorously? 

Reviewer #1: Yes

Reviewer #2: I Don't Know

Reviewer #3: Yes

Reviewer #4: Yes

3. Have the authors made all data underlying the findings in their manuscript fully available?

Reviewer #1: No

Reviewer #2: No

Reviewer #3: No

Reviewer #4: Yes

4. Is the manuscript presented in an intelligible fashion and written in standard English?

Reviewer #1: Yes

Reviewer #2: Yes

Reviewer #3: Yes

Reviewer #4: Yes

5. Review Comments to the Author

Reviewer #1: Thank you for inviting me to review this manuscript, which analyses the contributions of genes and paternal involvement to the intergenerational transmission of educational attainment. Data come from the Add Health Study of 5,021 genotyped adolescents and their parents. The study finds that both genes and father involvement (over and above mother involvement) are contribute to the intergenerational transmission of educational attainment; that these mechanisms are correlated with each other; and that part of the association of fathers’ school- specific involvement and educational attainment is potentially confounded by children’s education genetics.

Overall I thought this study was well-done and well-written. I particularly appreciated how the manuscript very clearly and systematically laid out and tested each hypothesis. I only have a few comments; most of these are about the language used when describing how different variables relate to each other. The study cannot establish causality, yet the language that is used often implies causality. This is problematic especially in genetics studies, because genetics studies are very prone to misinterpretation (which can have real-life devastating consequences, see recent events in the US). I used the following guide to replace terms in the paper, I may not have found every instance of the words used though and would ask the authors to carefully go through their manuscript and replace causal language with the appropriate wording:

shaping = is associated with

results in = is associated with

has an effect on = is associated with

impacts on = is associated with

leads to = is associated with

My list of suggested word-changes with page number is appended below; other comments I had were the following:

(1) For a general readership, it would be helpful to explain a bit more what a polygenic score is (i.e. explain what a GWAS does; emphasise that this is an aggregate score made up of many genetic variants, i.e. it is not a candidate gene study

(2) On Page 4 and 6 it says in the headlines for each hypothesis “independent” (e.g. “Genetic influences as an independent mechanism underlying the intergenerational transmission of educational attainment”) – it is not clear what the “independent” refers to (independent of what?) - would delete or clarify

(3) On Page 7 it says “children inherit half of their genes from each parent, and these parents also rear them and shape their environment.” it would be good to emphasise that this is only the case in biological families, which not all families are (e.g. sth like “children inherit half of their genes from each biological parent, and, if children live with their biological parents, these same parents also rear them and shape their environment)

(4) The distinction between direct and indirect effects (p9) is a little vague. Direct effects are genetic associations with an individual’s outcome that originate in that individual’s genetics; indirect genetic effects are associations that originate in another individual’s genetics (e.g. parents). In the current description it sounds as if direct genetic effects refer to effects that are mediated by a person’s behaviour, and indirect genetic effects are effects that are mediated by environments. However, this is incorrect, because effects of an individual’s genetics could be mediated by the environment (e.g. via evocative gene-environment correlation) yet they would still be direct effects (because they originate in that individual’s genetics). See also this paper:

Young, A. I., Benonisdottir, S., Przeworski, M., & Kong, A. (2019). Deconstructing the sources of genotype-phenotype associations in humans. Science, 365(6460), 1396-1400.

(5) The methods say “Non-European descent individuals were removed from the sample” – how was this done? Using self-identified ancestry? Genetically-identified ancestry?

(6) What are the estimates in Table 2 – standardised beta coefficients? Or unstandardised estimates? Please clarify.

(7) Authors interpret their findings to suggest that “that a substantial part of the role the father’s school-specific involvement plays in the intergenerational transmission of years of education is genetically confounded” (p 21), but the confounding is 10%. That doesn’t seem very substantial. Would rephrase accordingly.

(8) Another proposed robustness check: what happens, at least for the main analyses (as the sibling sample would probably end up too small), if the sample is restricted to those where children live with their father?

Here are my suggested changes of wording to replace language that implies causality (not these are suggested changes – happy for the authors to use their own wording, as long as it fixes the issue).

(9) page 6, change “40% of the variation in education can be explained by genetic variation” to “40% of the variation in education is associated with genetic variation”, because “explained” implies causality, and there is nothing causal about variance decomposition analysis. Likewise, page 6 “Children with genes that are positively related to higher educational attainment are more open”, change to “Children with genes that are positively related to higher educational attainment tend to be more open” or “On average, children with genes..” and then the second part of the sentence reads “..which all result in better educational achievements”, which again heavily implies causality and should be changed to sth like “which are linked with better educational achievements”.

(10) page 7 “the parent’s education PGS not only shapes their own educational attainment” should be sth like “the parent’s education PGS is not only associated with their own educational attainment”.

(11) page 8 “both fathers’ involvement as well children’s educational attainment is shaped by the same genetic factors” replace with “both fathers’ involvement as well children’s educational attainment is associated with the same genetic factors” ; same page “the same genes that result in higher education, also result in” should say “the same genes that are associated with higher education, are also associated with..”

(12) page 10: “do genes and father involvement independently explain the intergenerational transmission of education” change to sth like “are genes and father involvement independently associated with the intergenerational transmission of education” ; “to what extent do genes explain part of the behaviour mechanism” change to “to what extent do genes account for the behaviour mechanism” or “to what extent do genes confound the behaviour mechanism”

(13) page 11: “we can tap into the causes of the hypothesized correlation ” change to “we can test the hypothesized causes of any correlation ”

Reviewer #2: The manuscript “The intergenerational transmission of educational attainment: A closer look at the (interrelated) roles of paternal involvement and genetic inheritance” PONE-D-22-08657 investigates the joint effect of paternal involvement and education polygenic score (PGS) in predicting educational attainment, as well as their contribution as mechanisms of intergenerational transmission of education. The study is well motivated (with some reservations described below) to make a reasonable contribution to our understanding in an important topic. However, some empirical decisions and their presentation is confusing and the article overall requires polishing before the publication can be recommended.

I am admittingly not an expert in path modelling, but I struggle to understand the analysis description:

First, could the authors explain the intuition behind the multilevel models assessing rGE? As described on page 16, they fit model of something like:

Education=education_mother+education_father+PCs+controls+ζ+ε,

and then use ε (individual-level residual, I assume, although the authors do not specify which of the two residual terms ζ/ε, or even both, they use) of the model above to fit:

ε = PGS (or corresponding models examining two dimensions of father’s involvement)

What is the advantage of this approach compared to, for example, the more straightforward method of fitting first a model without mediators and then with mediators, and assessing the attenuation between these models?

Second, for hypothesis 3, why there is no need for other controls that PCs (page 17)?

Third, for the description of hypotheses 1, 2 ,4 and 5, it may be beneficial to state explicitly in what way the coefficients are compared. In addition, I would like to see some details on bootstrap simulations (method, has the multilevel structure been taken into account in sampling, how many replications).

Motivational and interpretational issues:

Authors motivate their focus on paternal involvement based on that “we do expect to see greater variation in paternal than in maternal involvement, and this is our main rationale for choosing to focus on paternal involvement in the current paper” (page 3). Based on Table 1, the difference in standard deviations between both dimensions of paternal and maternal involvement seems to be rather trivial. I tend to think that the paper might be stronger if both paternal and maternal involvement were on the focus, but I do not demand such change if authors think, for example, that this makes focus too scattered. Nevertheless, based on the evidence, I do not buy this specific argument for the current focus.

On page 24 authors state that “findings from the field of behavioural sciences have likely overestimated the role that fathers’ school-specific involvement plays in the intergenerational transmission of educational attainment.” Is this interpretation consistent with the results? Within-family analysis did not show any attenuations . Although subject of low power and thus only suggestive, wouldn’t this mean that the correlation may stem fully from active rGE. Would this mean that the causal direction would flow from pgs to father involvement, i.e. the involvement is not confounded by the PGS, but acts as one of the mechanisms via which PGS operates. Am I missing something?

On variables, and related issues:

Does controlling for enrolment in school involve a potential “bad control” problem? It can be a mediator (or even a collider) instead of a confounder given the analysis focus.

On page 15, authors claim to control for “the first 10 principal components (PCs).” PCs of what data? I think I can guess the answer, but scientific writing should not leave readers with guessing games.

Are there overlapping samples between GWAS and analysis data?

Could/should the genotyping chip be controlled in the models?

The relative importance of mediators is hard to assess, as they are all in the different scales. Could they be standardized to SD units?

There is a new generation PGS of education (Okbay et al. 2022, Nature genetics, 54(4), 437-449.). Could the new score be accommodated in the revision? However, the improvement is likely to be marginal, as the increase in sample size comes from 23andme, which usually cannot be shared. Thus, if this cannot be easily done, I understand if the authors want to skip this suggestion.

Reliability assessments of father involvement scales (Cronbach’s α or similar) would be nice.

Presentational issues:

• Standard errors (or Confidence intervals if authors prefer) could be presented in all tables & Figures. If not possible, then at least in the text when referring to estimates. There is also no need for vague description of p-values (e.g. “borderline significant” p.21) or asterisks when referring to them in the text. Precise values could be presented as easily.

• Figures 1-3 could be integrated into different panels of one figure.

• I prefer an old-school correlation matrix (with numbers) as Figure 4 relative to heat map

• Figure 5: It is hard to follow, where 0.016(ns) refers. I guess that between PGS and leisure involvement, but it took its time to understand (if correct).

• “Correlated effects” is as exotic term. What is this actually? The correlation, simply, or something else? Possibly clarifying the methods section may help here as well.

• Overall, it is very unconventional to present outcomes of regression models in the table rows as done in Table 2. This may cause misconceptions.

• There is no need to put different row on “N sibling pairs” in table 2. The old rows, “N individuals” and “N families” could accommodate also sibling design nicely

Issues (again mostly presentational) regarding the analyses of appendix

• Table S1 (upper panel). How can R2 drop between models 2 and 3?

• I would like to see direct effect in the lower panel of table S1

• Table S2 Contrasting OLS and FE models may be a category mistake. Linear FE models are also typically estimated via OLS. And even if not in this specific case, the essential substance-related difference is not the estimation method.

• P value can never be exactly 0

Reviewer #3: This study employed the National Longitudinal Study of Adolescent to Adult Health (Add Health) to provide a deeper understanding of the potential role of paternal involvement in intergenerational transmission in academic attainment. The results revealed that both genetic influences as well as father involvement effectively mediate the association between paternal and offspring academic attainment. I believe this study addresses an interesting topic, particularly its emphasis on paternal involvement, is generally well-written, and has the potential to make a meaningful contribution to the extant literature. With that said, however, there are a few areas that can be improved to more effectively display the underlying contribution and further inform future research. I’ve provided a summary of these areas below with some suggestions for the authors to consider. Best of luck with your revisions and thank you for the opportunity to review your work.

On page 2, the authors, rightfully, point out that previous studies have revealed that the association between family environment and educational attainment may be artificially inflated in light of shared genetic influences passed from parents to offspring that collectively contribute to both increased genetic predisposition for variation in educational attainment as well as the environments that parents design for their children. The latter is, at least to some degree, also a reflection of parent predisposition toward educational attainment—and related phenotypes. The resulting covariation between genetic predisposition and these specialized environments that explain variance in educational attainment is, as the authors note, an example of genetic confounding (as well as a passive rGE more specifically). All of this is to say that I completely agree with the authors assessment of this limitation in the literature, but I think it would be beneficial to expand on the underlying meaning of “genetic confounding” in this context as some readers may not be as familiar with this concept and the theoretical and methodological problems that it may give rise to.

Similar to my previous comment, the authors summarize Wertz et al.’s (2020) findings on pages 2-3 of the manuscript. Again, the authors provide a sufficient and accurate description of the concept of “genetic nurture” but I think a slightly more expanded definition and description of the supposed underlying mechanisms underlying genetic nurture would be beneficial for readers that are either unfamiliar with this concept or who are trying to understand how it applies to the current study more directly.

I think the authors do a great job of setting up their arguments for shifting focus to paternal involvement within the context of the current study throughout the literature review; however, once they reach the penultimate paragraph (the final full paragraph on page 4) I believe the authors can be a bit more direct. They mention that they are examining paternal involvement and genes (from a GWAS), but they do not provide any indication of how they will examine these two sources of influence. Will the GWAS measure simply serve as a control? Will they examine gene-environment interplay? Again, just a couple of sentences here to flesh things out a bit more may provide readers with valuable information regarding the primary goals of the study.

Hypothesis 1 frame paternal involvement as a mechanism of intergenerational educational attainment. Based on the arguments offered by the authors, I believe this is a reasonable hypothesis. With that said, do the authors believe it is at least possible that at least some of the covariation between paternal educational attainment and involvement is the result of a set of a single suite of genetic influences (or related genetic influences)? I think it is at least possible that educational attainment and parental involvement may be the result of shared genetic influences operating on higher order phenotypes (e.g., impulsivity), of which educational attainment and involvement may reflect more proximately. This could be addressed methodologically with paternal GWAS scores for educational attainment (or involvement, I suppose), but I think the authors need to at least explore/discuss this possibility more directly.

The addition of maternal involvement and educational attainment into the multivariate equation significantly strengthens the estimated models. Given the estimated indirect effects, however, it is not currently clear how these measures were “controlled.” In other words, did the authors regress the examined outcome (child educational attainment), mediators (father involvement and the PGS), and primary IV (father educational attainment) on the examined controls or just a subset of these measures?

The extent to which the examined PGS mediated the association between parental and offspring educational attainment was extremely interesting, in my opinion. The authors briefly discuss these findings on pages 22-23, but I think some additional expansion would be useful. More specifically, there has been much discussion surrounding the utility of PGSs as of late with many critics (perhaps, rightfully) challenging the notion of genetic influences as a source of causality. I’m not suggesting the authors tread into these choppy waters, but I do think that framing a PGS as a potential mechanism rather than a source of causal influence may be beneficial given their findings. We are still trying to figure out exactly what the variance explained by a PGS is and how to best leverage these measures. I believe the authors’ findings may provide some additional and useful insight in that perhaps we are better suited examining PGSs as a source of intergenerational transmission (when appropriate) rather than a source of more general causality. I don’t think the authors need to go too far down this rabbit hole, but some additional expansion here would be beneficial for future research in this area and also highlights an additional contribution of the current study to the extant literature.

Reviewer #4: This is an overall well-written paper exploring the mechanisms explaining the intergenerational transmission of educational attainment focusing on paternal involvement. While we know little about how intergenerational transmission works, and know little about maternal influences on educational outcomes, we know even less about paternal involvement in explaining educational outcomes in offspring. The paper adds to this major research gap. The manuscript is methodologically sound and suited for publication in PlosOne. The topic is very important and of interest to researchers from varied disciplines as well as for policymakers and will hopefully spark more research into intergenerational transmission of educational attainment using genetically sensitive designs. I have some minor concerns for the authors to consider.

• “To obtain a more complete understanding, the current study integrates insights from the fields of behavioral sciences and genetics and examines the extent to which factors from each field are unique underlying mechanisms, correlate with each other, and/or act as important confounders in the intergenerational transmission of educational attainment.” � this to me seemed like the study was looking into several mechanisms, that would be parental, grandparental, sibling, and societal effects on offspring/sibling outcomes, etc. I suggest rephrasing the sentence in the abstract to reflect more precisely what the study was looking into. Paternal effects are grossly understudied, so it is a very valuable study on its own.

• I suggest adding effect sizes to the abstract. That is, before talking about mediation analyses, state the direct effect, what is the effect size of the correlations between behavioral and genetic influences, etc.

• Nicely written introduction. I suggest including that SES itself is partly explained by genetic factors, while often assumed to be environmental. It would also be helpful if effect sizes are including in the introduction, for example, the magnitude of correlations between paternal teaching-related activities and offspring educational attainment, etc. (or for example, Children with genes that are positively related to higher educational attainment are more open, agreeable, conscientious, and show more academic motivation, which all result in better educational achievements – what are the effect sizes here?).

• A nice addition would have been to include data about parental genotypes- perhaps this could be discussed in the paper?

• Could you please unpack this? “PGSs cannot distinguish between “direct” genetic effects-associations between genes and education through intelligence and motivation” - What is the direct genetic effect? The genetic variants are not coding for educational outcomes, not even through intelligence and/or motivation. Same here: “and indirect genetic effects -associations between genes and education due to the family environment and parenting practices” Do the authors mean genetic factors explaining family environment and parenting?

• I suggest discussing the representativeness of the sample. Is the data missing at random? Does the genotyped sample reduce the representativeness? How about information available about paternal involvement? Some sensitivity analysis would be useful.

• The methodology is sound, however, I suggest talking about effect sizes rather than significance, e.g., “The effect of the father’s school-specific involvement is significantly reduced from 0.056 to 0.050 when including the child’s education PGS, which implies 10.7% genetic confounding, while the effect of the father’s leisure involvement is not significantly reduced when including children’s education PGS.’ � this is an interesting but a small effect, it is significant because of the large sample size. It is important to note this.

• Was the analysis plan preregistered? How was multiple testing controlled for?

• “Comparing the between-families and within-families associations between father involvement and children’s education PGS provides insights into the extent to which the rGE is active or passive.” � I do not think you can distinguish between active and evocative rGE?

• I suggest not using the term “borderline significant’ it is either significant or not significant. It is especially questionable to interpret the results as showing effect or ‘hinting’ to an effect.

• It is commendable that several robustness checks were done. I suggest doing some checks about missing data and representativeness as well.

• I suggest adding precise figure legends, what is presented, what are the error terms in parentheses (e.g. figure 5)

6. PLOS authors have the option to publish the peer review history of their article (what does this mean?). If published, this will include your full peer review and any attached files.

Reviewer #1: No

Reviewer #2: No

Reviewer #3: No

Reviewer #4: No

---

## [Author Response · Author response to Decision Letter 0]

31 Oct 2022

Please see attached document with all responses

---

## [Decision Letter · Decision Letter 1]

23 Nov 2022

The intergenerational transmission of educational attainment: A closer look at the (interrelated) roles of paternal involvement and genetic inheritance

PONE-D-22-08657R1

Dear Dr. Verweij,

We’re pleased to inform you that your manuscript has been judged scientifically suitable for publication and will be formally accepted for publication once it meets all outstanding technical requirements.

Kind regards,

Marie-Pierre Dubé, PhD

Academic Editor

PLOS ONE

Additional Editor Comments (optional):

Reviewers' comments:

Reviewer's Responses to Questions

**Comments to the Author**

1. If the authors have adequately addressed your comments raised in a previous round of review and you feel that this manuscript is now acceptable for publication, you may indicate that here to bypass the “Comments to the Author” section, enter your conflict of interest statement in the “Confidential to Editor” section, and submit your "Accept" recommendation.

Reviewer #1: All comments have been addressed

Reviewer #2: All comments have been addressed

Reviewer #3: All comments have been addressed

Reviewer #4: All comments have been addressed

2. Is the manuscript technically sound, and do the data support the conclusions?

Reviewer #1: Yes

Reviewer #2: Yes

Reviewer #3: Yes

Reviewer #4: Yes

3. Has the statistical analysis been performed appropriately and rigorously? 

Reviewer #1: Yes

Reviewer #2: Yes

Reviewer #3: Yes

Reviewer #4: Yes

4. Have the authors made all data underlying the findings in their manuscript fully available?

Reviewer #1: Yes

Reviewer #2: No

Reviewer #3: No

Reviewer #4: (No Response)

5. Is the manuscript presented in an intelligible fashion and written in standard English?

Reviewer #1: Yes

Reviewer #2: Yes

Reviewer #3: Yes

Reviewer #4: Yes

6. Review Comments to the Author

Reviewer #1: The authors have taken great care in responding to my comments, which I appreciate. I'm satisfied with their revisions.

Reviewer #2: The authors have provided reasonable answers to my concerns in the first round. I have no further requests, and recommend the publication of the study!

Reviewer #3: The authors have adequately addressed all of my concerns. I see no reason the study cannot be published in its current form.

Reviewer #4: Thank you for addressing all the reviewer comments so carefully. I am generally very happy with the revision, although I do not agree with your decision about not correcting for multiple testing, especially as the analyses plan was not preregistered. At the very least, I suggest adding the justification for the decision to the manuscript.

7. PLOS authors have the option to publish the peer review history of their article (what does this mean?). If published, this will include your full peer review and any attached files.

Reviewer #1: **Yes: **Jasmin Wertz

Reviewer #2: No

Reviewer #3: No

Reviewer #4: No

---

## [Editor Report · Acceptance letter]

1 Dec 2022

PONE-D-22-08657R1 

The intergenerational transmission of educational attainment: A closer look at the (interrelated) roles of paternal involvement and genetic inheritance 

Dear Dr. Verweij:

I'm pleased to inform you that your manuscript has been deemed suitable for publication in PLOS ONE. Congratulations! Your manuscript is now with our production department. 

Kind regards, 

on behalf of

Dr. Marie-Pierre Dubé 

Academic Editor

PLOS ONE